# Can Plants Move Like Animals? A Three-Dimensional Stereovision Analysis of Movement in Plants

**DOI:** 10.3390/ani11071854

**Published:** 2021-06-22

**Authors:** Valentina Simonetti, Maria Bulgheroni, Silvia Guerra, Alessandro Peressotti, Francesca Peressotti, Walter Baccinelli, Francesco Ceccarini, Bianca Bonato, Qiuran Wang, Umberto Castiello

**Affiliations:** 1Ab.Acus srl, 20155 Milan, Italy; mariabulgheroni@ab-acus.com (M.B.); walterbaccinelli@ab-acus.eu (W.B.); 2Dipartimento di Psicologia Generale, Università di Padova, 35131 Padova, Italy; silvia.guerra.5@phd.unipd.it (S.G.); ceccarini1980@gmail.com (F.C.); bianca.bonato.1@phd.unipd.it (B.B.); qiuran.wang@phd.unipd.it (Q.W.); 3Dipartimento di Scienze Agroalimentari, Ambientali e Animali, Università degli Studi di Udine, 33100 Udine, Italy; alessandro.peressotti@uniud.it; 4Dipartimento di Psicologia dello Sviluppo e della Socializzazione, Università degli Studi di Padova, 35131 Padova, Italy; francesca.peressotti@unipd.it

**Keywords:** kinematics, circumnutation, plant behavior, plant movement

## Abstract

**Simple Summary:**

Intrigued by the ability of climbing peas to detect and grasp structures such as garden reeds, we adapted a method classically used to investigate the grasping movement of animals to the study of grasping movements in plants. We used time-lapse photography to document the behavior of pea plants, grown in the vicinity of a support pole. Using this footage, we analyzed the kinematics of tendrils growth and found that their approach and grasp exhibited movement signatures comparable to those characterizing the reach-to-grasp movement of animals. Through our method it may be possible to demonstrate that plants may be more sentient than we give them credit for: namely, they may possess the ability to act intentionally.

**Abstract:**

In this article we adapt a methodology customarily used to investigate movement in animals to study the movement of plants. The targeted movement is circumnutation, a helical organ movement widespread among plants. It is variable due to a different magnitude of the trajectory (amplitude) exhibited by the organ tip, duration of one cycle (period), circular, elliptical, pendulum-like or irregular shape and the clockwise and counterclockwise direction of rotation. The acquisition setup consists of two cameras used to obtain a stereoscopic vision for each plant. Cameras switch to infrared recording mode for low light level conditions, allowing continuous motion acquisition during the night. A dedicated software enables semi-automatic tracking of key points of the plant and reconstructs the 3D trajectory of each point along the whole movement. Three-dimensional trajectories for different points undergo a specific processing to compute those features suitable to describe circumnutation (e.g., maximum speed, circumnutation center, circumnutation length, etc.). By applying our method to the approach-to-grasp movement exhibited by climbing plants (*Pisum sativum* L.) it appears clear that the plants scale movement kinematics according to the features of the support in ways that are adaptive, flexible, anticipatory and goal-directed, reminiscent of how animals would act.

## 1. Introduction

Although terrestrial plants are unable to move from one place to another, they are very much in tune with their environment and are very capable of a variety of movements. Plants may not move as far or as quickly as animals, but some of their movements achieve many of the same functional ends as those of animals [1,2]. Here we focus on circumnutation, a helical movement of elongating plant organs that has been investigated for decades [3]. Changes in circumnutation patterns can provide useful insights for auto-ecological and physiological studies (e.g., [4]), and contribute to explaining the mechanisms that drive gravitropism [5,6,7], circadian rhythms [8,9,10] and growth rate [11]. Despite its importance as a plant physiological signature, the standard approach for circumnutation analysis still needs to be consolidated. Some studies make use of time-lapse images and adopt video processing to extract movement-related features [5,6,8,10,12,13,14,15]. For instance, Stolarz and colleagues [16] described a structured approach for 2D plant circumnutation analysis and developed a software (i.e., Circumnutation Tracker) for the extraction of key parameters with a standard set up that includes time-lapse video acquisition of the plant from a top view of the circumnutation movement, manual harvesting of the coordinates and parameters calculation. Their time-lapse video acquisition works on standard video formats, but it does not take into account possible lens distortion of the cameras that may introduce discrepancies between the real movement and recorded movement. A top view positioning of the camera has been adopted in a variety of studies [5,10,14,17], but it determines constraints on camera position and the type of points that can be investigated (points with circumnutating movement along with the camera view). A two-dimensional top view also limits the trajectory analysis of the camera’s field of view, precluding the recording of movements in other directions. In this respect, some studies tried to improve this set up by adding side view contextual acquisition [5,14,17]. As manual harvesting of coordinates can become a heavy process with the increase of video duration and points to track, Stolarz and colleagues [16] considered the automatization of this process as a next step for a future development of their system.

Johnsonn et al. [13] described a possible approach for 3D movement reconstruction with a system that records stereovision pictures by rotating two mirrors with a control mechanism. Their system is still under development and its use is strongly linked to the hardware, which has been built to meet the specific needs of their experimental setup, making its adaptation to other experimental conditions difficult.

More recently, Raja et al. [15] have proposed a minimalistic 1D approach that relies on nonlinear methods of behavioral analysis to uncover the dynamics of plant nutation. This approach focuses on the time dependencies characterizing the processes that give rise to circumnutation patterns and may help for the identification of alternative parameters than those derived from classic kinematics.

Continuing on this analysis, Bastien and colleagues ([18]; see also Porat et al. [19], Gallentine et al. [20]) developed computational 3D models to emulate growth dynamics in rod-like organs, including the effect of external stimuli in the environment. They focused on the limits of experimental measurements of nutation that use the projection of the apical part of the organ in the plane orthogonal to the gravitational field. They show that a complete kinematic description should consider geometrical and local effects in the 3D space and should not restrict the analysis to the position of the apical tip. They also stress the importance of future development of measurements considering proper 3D kinematics.

Here, we present an easy-to-scale system solution that: (1) leverages a pair of commercial cameras for 3D plant movement reconstruction; (2) compensates for possible lens distortion artifacts allowing a more robust computation of coordinates; and (3) implements semi-automatic trajectories that harvest and extract the circumnutation parameters. This approach is adaptable to different plants and experimental setups, which permits the investigation of motion in plants and allows for a comparison with the kinematical ways used to investigate movements in animal species.

## 2. Materials and Methods

The solution proposed in this paper is based on the acquisition and analysis of the plant’s movement through a stereovision system that generates time-lapses of the plant and reconstructs its 3D movement. The system workflow (Figure 1) envisages the acquisition of temporally equally spaced images of the plant by two fixed calibrated cameras. A semi-automatic point tracking process is applied to reconstruct the 3D trajectory of custom landmarks over the entire acquisition. Finally, the descriptive features of the 3D movement of the plant are computed from the analysis of the trajectories. The following sections illustrate the process details. The system has been extensively used for a specific experimental setup designed to investigate circumnutation pattern of climbing peas (*Pisum sativum* L. var. saccharatum cv Carouby de Maussane; hereafter *P. sativum*)

### 2.1. Instrumentation

A pair of RGB-infrared cameras has to be set for each plant under investigation. In our experimental setup we installed two cameras (i.e., IP 2.1 Mpx outdoor varifocal IR 1080 P) inside each thermo-light-controlled growth chamber. Both cameras were placed on the same side of the chamber to obtain a complete view of the organ to track, thus enabling a stereovision image recording. The cameras were set to acquire standard RGB images in high light conditions and automatically switch to infrared recording mode in low light condition. Each camera was wired with Ethernet cables to a wireless router (D-link Dsr-250n, D-Link Corporation Ltd., Taiwan) connected via Wi-Fi to a PC used as a data storage unit. A dedicated software, CamRecorder (Ab.Acus s.r.l. Milan, Italy), was developed to synchronize acquisition and customize frame rate (0.0056 Hz for the experimental protocols tested) for each pair of cameras.

### 2.2. Cameras Calibration and Stereovision

We developed a Matlab procedure (Camera Calibrator app) to reduce the cameras’ optical lens distortions and avoid geometrical position errors [21,22,23,24]. This procedure is run for each of the two lenses by ingesting 20 pictures of a paper-printed chessboard (squares’ side 18 mm, 10 columns, 7 rows) from multiple angles and distances, in natural, non-direct light conditions. The reprojection error is evaluated and images with high mean residual error (>1 pixel) are discarded and substituted with new pictures. This process is repeated until a set of 20 images with a low reprojection error is obtained. The lens distortion parameters computed for each camera are used to compensate the effects on the acquired images.

In addition to the aforementioned process for the computation of cameras’ intrinsic parameters, the extrinsic parameters for each pair of cameras have to be computed as to obtain the cameras’ relative positions. From the 2D positions of the images acquired by the two cameras, the 3D position for each point is reconstructed. The cameras’ relative positions were calculated with the same Matlab procedure from simultaneous pictures of a chessboard target with known landmarks. The target was placed in the middle of the growing chamber before plant growing onset.

### 2.3. Video Processing

The dedicated software SPROUTS (Ab.Acus s.r.l. Milan, Italy) was developed by using Python 3.7 to enable a user-friendly tracking for the considered key landmarks. The software is designed to work for any kind of growing setup that provides image streams from two cameras. Through a simple interface users can: (1) perform a semi-automatic tracking of custom number of key points, (2) compute 3D trajectories of each tracked point in real word dimensions and (3) save them as .cvs files containing the coordinates of the point in each frame. Initially, the user is asked to identify the points to be tracked on the first image, then a tracking algorithm estimates the position of the points on the following frames. The user is allowed to check for position estimation errors eventually introduced by the automatic tracking procedure: the user can re-mark the point to be tracked on the first available image, supervise the tracking process and eventually adjust the trajectory of the point being tracked. At the end of the tracking process, the user can review the position of the tracked points in all the images and eventually correct the positioning errors. Such semi-automatic tracking is implemented using the Lucas Kanade computer vision method [25] based on optical flow, using a size of the search window of 10 by 10 pixels, a maximal pyramid level number (iterative lowering of image resolution) equal to 20 and the termination criteria of 30 maximum iterations. Three-dimensional trajectories are obtained from 2D trajectories acquired for both the left and right-side cameras using the Matlab triangulating function [26]. A sample for the 2D trajectory extracted for one camera is shown in the Appendix A, whereas an example of the reconstructed 3D trajectory is shown in Appendix A.

### 2.4. Features Extraction

Finally, the 2D and 3D positions and kinematic features are calculated from coordinates of landmarks with a dedicated procedure. The extraction module was developed in Matlab 2020a and was designed to process the 3D trajectory files obtained from the previously introduced Video Processing steps. The 3D point position reconstruction algorithms provide the x, y and z coordinates of each point of the trajectory in a reference system with its origin on the lower left corner of the calibration chessboard. Since the positioning of the chessboard cannot be consistent on all the experiments, to enable the extraction computation of the correct features, the points’ reference system is roto-translated to the plant’s reference system.

The new reference system is built such that the system origin coincides with one of the points along the vertical axis of the plant not showing relevant movement during the acquisition. The y-axis of the new 3D system corresponds to the plant’s vertical axis, while the x and z axes lay on the plane perpendicular to the y-axis containing the system origin. To correctly identify the plant’s vertical axis and origin, the user is required to position two points along the plant’s axis during the video processing. The vertical axis is the line passing through these two points. In the case of pea plants (*P. sativum*), the y-axis was set as the line passing through the first and second internode, with the system origin being the first internode (Figure 2).

Once the reference system is aligned to the plant’s system, the software detects circumnutating movements. Taking the X–Z components of the 3D movement, the software computes the angle (α) between the horizontal axis (x-axis) and the movement vector of each frame, considered as the vector between the tracked point at frame t (P2) and the same point at frame t-1 (P1) as shown in Figure 3a. Figure 3b shows the result of the whole trajectory, with a five-sample moving average. A single circumnutation movement is considered as the movement occurring between two local maxima of the α angle. Figure 4 shows the result of the extraction of all the circumnutations from the trajectory obtained from a *P. sativum* in 35 h of movement.

After the circumnutations segmentation, a set of features aimed at describing the movement of plants is computed. Such features can be classified in four different functional categories: circumnutation related features, whole movement related features, point related features and experimental specific features. Along with the features provided by the literature [8,11,12,16], we enlarged the panorama of measures available with extra features representative of the movement of the plant. Below are the details on each computed feature.

### 2.5. Circumnutation Related Features

Circumnutation related features provide a quantitative description of each circumnutation extracted from the movement segmentation. Along with the metrics related to circumnutation’s kinematics, the algorithm also computes indices related to circumnutation’s shape and orientation. Such features are:Duration: seconds needed to complete the whole circumnutation movement. It is computed as the time between two local maxima of the α angle shown in Figure 3a.Mean/Max/Min Circumnutation Speed: values for mean value, maximum value and minimum value for the speed traveled by the circumnutating point along each circumnutation.Circumnutation Path Length: the overall 3D path travelled by the circumnutating point. Computed as the sum of all the Euclidean distances between subsequent point positions during the single circumnutation:
(1)$$Circumnutation\;Path\;Length=\overset{T-1\;}{\underset{t=1}{\sum }}\sqrt{{\left(x\left(t\right)\;-\;x\left(t+1\right)\right)}^{2}+{\left(y\left(t\right)\;-\;y\left(t+1\right)\right)}^{2}+{\left(z\left(t\right)\;-\;z\left(t+1\right)\right)}^{2}}$$
where:

T: total number of points in the circumnutation; $x\left(t\right),y(t),z(t)$: point 3D coordinates at time t; $x\left(t+1\right),y\left(t+1\right),z(t+1)$: point 3D coordinates at time *t* + 1

4.Circumnutation Center: geometric center of gravity in the X–Z plane computed as the mean of each coordinate for all the points constituting the circumnutation.5.Circumnutation Center Distance from Plant origin: euclidean distance between the Circumnutation Center and the plant origin in the X–Z plane.6.Circumnutation Center Speed: speed of the Circumnutation Center computed as the distance traveled by the Circumnutation Center in two consecutive frames on the X–Z plane divided by the time between frames.7.Circumnutation Centroid: considering the points of the trajectory described by the circumnutation on the X–Z plane, the algorithm identifies the region of interest as the closed line obtained by linking such points. The pixels in the polygon area are set to 1, while the others are set to 0. The Circumnutation Centroid is calculated as the geometric center of gravity in the X–Z plane computed as the mean of each coordinate for all the points with value 1 contained in the circumnutation area.8.Circumnutation Centroid Distance from Plant origin: the Euclidean distance between the Circumnutation Centroid and the plant origin in the X–Z plane.9.*Circumnutation Centroid Speed:* speed of the Circumnutation Centroid computed as the distance traveled by the Circumnutation Centroid in two consecutive frames on the X–Z plane divided by the time between frames.10.Circumnutation Main Axis: the principal axis of the ellipsoid of the circumnutation, computed as the maximum distance between two points of the circumnutation trajectory in the X–Z plane.11.Circumnutation Area: the sum of pixels with a value equal to 1 obtained from the binarization of the circumnutation trajectory on the X–Z plane as described for the calculation of Circumnutation Centroid.12.Direction: clockwise or counterclockwise. For each circumnutation, the software computes the sum of all the angles between the movement vector at time *t* and the movement vector at time t+1. The direction, then, is determined according to the following logic: if the resulting sum is equal to 2π ± 1.2, then the direction is counterclockwise, or else if the resulting sum is equal to −2π ± 1.2, then the direction is clockwise. For all other cases, no direction is assigned. The pseudocode shown below describes the direction estimation logic.

***if:*** $\sum_{\mathrm{movementsC}}\;angleBetween\left(m\left(t\right),m\left(t+1\right)\right)\;=2\pi \;\pm \;1.2$ ***then:** direction = “counter – clockwise”****else if:*** $\sum_{\mathrm{movementsC}}\;angleBetween\left(m\left(t\right),m\left(t+1\right)\right)\;=\;-\left(2\pi \;\pm \;1.2\right)$ ***then:** direction = “clockwise”****else**:direction = “None”*
where:

movementsC: set of all movement vectors in a single circumnutation; $m\left(t\right)$: movement vector at time t; $m\left(t+1\right)$: movement vector at time *t* + 1.

### 2.6. Whole Movement Related Features

Whole movement related features provide descriptions of the plant’s complete movement, from circumnutation onset to movement end. Such features are:Max Circumnutation—Circumnutation Path Length: maximum value of the 3D Circumnutation Path Length for all the circumnutations. Example shown in Figure 5.

2.Max Circumnutation–Circumnutation area: maximum value of Circumnutation Area in the X–Z plane for all the circumnutations.3.Direction switches: sum of all direction switches in circumnutations along the whole movement. Example shown in Figure 6.

### 2.7. Point Related Features

Point related features are metrics related to circumnutating point kinematics along the movement. Such features are:Mean and Maximum Speed: mean and maximum speed of the point reached along with the whole movement. Speed is computed as the distance traveled by the point between consecutive frames divided by the time between frames.Time of maximum speed: the time at which maximum speed is reached both as absolute time and as a percentage with respect to the whole movement.

### 2.8. Experimental-Specific Features

The system was used intensively on the climbing pea plant (*P. sativum*) to quantify its circumnutating movement toward a support (e.g., a stimulus). To exploit aspects related to this experimental setup the software was evolved to compute extra experimental specific features.

In details:Center/Centroid distance from stimulus: the Euclidean distance between Circumnutation Center/Centroid and the stimulus in the X–Z plane.Angle between circumnutation main axis and plant stimulus axis: the angle between Circumnutation Main Axis and plant-stimulus axis computed as the line passing through the plant origin and the stimulus on the X–Z axis.Minimum distance from stimulus: the 3D Euclidean distance between the stimulus and its closest point of the circumnutation trajectory.

## 3. Testing the Methodology

The complete system has been tested by different operators on a population of 49 snow peas (*P. sativum*) in eight individual growth chambers equipped with eight different camera couples without major issues reported.

A possible shortcoming of the system is the manual positioning of the points’ positions, both in the first frame and when positioning corrections are required.

Indeed, this manual intervention could result in an operator-dependent effect on the data. In order to verify the independence of measures extracted from the operator, two different users have been asked to perform a complete tracking of a circumnutating point on the same timelapse of a growing chamber and processing the same circumnutating point in order to compare trajectories obtained and the impact of operator intervention on features extracted. The two 3D trajectories obtained have been compared to check that the procedure was stable across users. For each tracked frame, the distance between the point’s position obtained by the two operators has been computed. The histogram of differences between the two trajectories is shown in Figure 7. The main differences observed are in the range between 0 and 5 mm with 90% of the points tracked showing less than 5 mm difference.

The correlation coefficients were computed between the trajectories obtained by the two users. Values of the correlation coefficients are reported in Table 1.

The level of agreement between the two operators on circumnutation related features has been evaluated computing Intraclass Correlation Coefficient (ICC) between features extracted from the first user and the second user. The ICC has been computed using the absolute agreement of the two-way random effect model following [27]. Results obtained on 29 circumnutations are shown in Table 2.

In addition to the computation of the ICC, a hypothesis test has been performed with the null hypothesis that ICC = 0, and all features considered scored a *p*-Value < 0.05.

From the ICC values obtained, all the features, except for center_distance_from_origin and centroid_distance_from_origin, were scored higher than 0.75, showing an excellent correlation. Centroid_distance_from_origin scored a 0.72 ICC value, showing good correlation, and center_distance_from_origin scored a 0.55 ICC value, showing fair correlation. In all cases, the *p*-Value obtained was <0.05, reporting the significance of values obtained. Overall, the analysis of the manual interventions impact revealed that the system results are stable across different users, indicating the reliability of the data obtained through the proposed methodology.

## 4. Discussion

Although the processes by which circumnutation occurs are well understood, the controlling mechanism underlying circumnutation is still unknown [8]. Informative content of circumnutation plays a key role in the study of the behavior of plants, hence the importance of having tools to ease and standardize the extraction and processing of circumnutation’s key features in 3D space can be crucial. Highly variable characteristics between different plant species require a flexible method to allow wider applicability of a standard approach to different plant structures and moving behaviors. The system proposed in this article addresses this need and makes a step forward to define a standard approach for 3D movement extraction and analysis in plants. An omni-comprehensive approach to the problem has been proposed and implemented from the data acquisition protocol and instrumentation to the data extraction and the features processing software and methodologies. A set of descriptors for the movement of plants has been proposed, starting from the circumnutation concept, with the aim of providing a quantification of the motion behavior of the plants. Some of these features have been already proposed and validated in previous studies, and a step forward was taken in the approach proposed in this paper, by extending their applicability from 2D to 3D space. In addition to this set of features, new descriptors inspired by the circumnutating movement characteristics and the standard kinematic analysis, have been proposed and implemented. Even though the system allows for a 3D reconstruction in time of custom selected points, the descriptive features extracted so far mainly focuses on the kinematic description of the organ tip, that may give an incomplete picture of motion (as also suggested by Bastien et al., [18]). We feel that a precise 3D reconstruction of different key points as identified on the plant may provide a more precise tool for the validation of computational 3D models [18,19]. A future development of our system should focus on the relative movement of selected points, the first approximation of the overall 3D structure or on a 3D surface reconstruction of the organ. Recently, the kinematical approach for the study of movements in plants has been disputed. Raja et al. [15] criticize what they name “kinematics of nutation” approaches, supporting the thesis that kinematical patterns are blind to temporal dependencies versus what they define a “dynamical methodology” for the study of plants dynamic of nutation. Despite the term “dynamics”, they do not consider a model of forces that drives the movement, as used in classical mechanics, but mostly focus on the temporal evolution of kinematic variables and the analysis of time series. They compute other kinematics-based descriptive features of behavioural dynamics such as harmonicity, predictability and complexity. Raja’s criticism builds on the interpretation of kinematics approaches as methodologies that merely provide movement patterns through the averaging of features along plant’s movement, supposing the absence of temporal dependencies. Overcoming the aforementioned interpretation, in this paper we have clarified that kinematic approaches also address the evolution of the movements of plants across time, building upon the computation and extraction of time series. Furthermore, some inherent limitations are present in the approach proposed by Raja et al. [15], where the information reduction through a single-point and single-coordinate analysis do not allow an evaluation of the behavior of different plant’s segments and their interrelation for the movement production. Oppositely, we show that the analysis of standard kinematics measures in a multi-point evaluation can reveal specific behavioural patterns [2]. In that study, the focus was on the relative movements of multiple tendrils belonging to the same circumnutating organ in climbing plants, where zoomorphic references (e.g., wrist, digits) were adopted with the intent of ease the understanding of the experimental conditions and to make a direct comparison with a well-understood animal movement [28].

To elaborate, Charles Darwin and his son Francis [3] observed that the tendrils of climbing plants during circumnutation tend to assume the shape of whatever surface they come into contact with; that is, they learn progressively the shape of potential support characteristics [29]. Implicitly this signifies that they perceive the support and plan the movement accordingly. In this view climbing plants might represent actions in terms of their perceivable consequences. This is a strong inference because studying the interaction among organisms and objects is grounded on the requirement of a central nervous system (CNS). What the observations collected by the Darwins suggest is that other options are available, and that they do not require a CNS for adaptive perceptuomotor transformations to be happening. Tendrils can develop in different forms with some of them resembling a kind of digit characterized as changes in aperture as the bending towards a potential support progresses. This kind of reach-to-grasp behaviour resembles that exhibited by human and non-human primates [28] as well as by other animal species such as tetrapods [30,31,32], which have also evolved significant forelimb prehensile capabilities. In all cases during the course of a reach-to-grasp movement, there is first a progressive opening of the appendage, followed by a gradual closure of the appendage until it matches the to-be-grasped object [33]. Research on reach-to-grasp kinematics has proven insightful in revealing how specific kinematic landmarks modulate with respect to object properties, including, for instance, object thickness [28]. Results which hold across animal species are that the velocity of hand opening during reaching is lower and the maximum aperture of the hand is smaller for thinner than for thicker stimuli. In this respect, ‘thickness’ offers an ideal opportunity to parallel the ways of grasping in animal and climbing plants given that the success of grasping a support by a climber heavily depends on the support’s diameter [34]. Differences in support thickness can determine changes in energy expenditure, which are visible on parameters characterizing the support-finding process [35]. With this in mind, we used our method to ascertain whether plants of *P. sativum* have the ability to perceive a potential support in the environment and modulate the kinematics of movement of the tendrils according to its thickness during the approach phase. The question is whether they are endowed with a purposeful behavior that is anticipatory in nature. We reasoned that if the principles of motion planning at the basis of animals’ and plants’ approach-to-grasp behavior are based on similar basic mechanisms, then the intrinsic properties of a support such as its thickness might have considerable effects on the kinematics of tendrils’ aperture during the approach-to-grasp behavior. The results speak clearly. Not only did the plants acknowledged the presence of the support, but they exhibited a different kinematic patterning depending on stimulus thickness ([2]; see also [36,37]; see Appendix A).

As shown in Figure 8, the average tendril’s velocity was significantly greater for the thin than for the large support, and the maximum velocity was significantly higher for the thin than for the thick stimulus. The time at which the tendrils reached the maximum velocity, calculated as a percentage of total movement time, was significantly earlier for the thick than for the thin support. The maximum distance between the tendrils was scaled with respect to the size of the support. It was significantly greater for the thin than for the large support. In addition, the time at which the tendrils reached the maximum aperture calculated as a percentage of the total movement time occurred significantly earlier for the thick than for the thin support. Based on these findings, the plants appear to behave in ways that are adaptive, flexible, anticipatory and goal-directed similar to how animals do. During the movement towards the support the plants scale, the maximum aperture and the velocity of the tendrils’ opening with respect to the thickness of the support increased. This evidence suggests that plants are able to process the properties of the support and are endowed with a form of perception underwriting a goal-directed and anticipatory behavior (Guerra et al., [2]). A caveat of these findings, however, is that our results indicate an opposite pattern of that reported in previous animal literature (e.g., Castiello & Dadda, [28]). Remember that the reach-to-grasp in animals demonstrates consistency across studies with regard to results such as a lower maximum peak velocity and an earlier and smaller maximum hand aperture for smaller stimuli relative to larger stimuli [28]. In general, this pattern has been explained in the terms that smaller stimuli require more accuracy and therefore lowering down the velocity allows for dealing with accuracy requirements.

A possible explanation for this discrepancy may reside in the fact that for plants reaching to grasp thick supports is a more energy consuming process than grasping thinner ones. Indeed, the grasping of a thick support implies that plants have to increase the tendrils’ length in order to efficiently coil the support [38] and to strengthen the tensional forces to resist gravity [35]. Since these processes are characterized by a high Adenosine triphosphate (ATP) consumption, coiling thicker supports results in more energy expenditure [34]. Coherently, the reduction of movement velocity during the approaching maneuver may allow climbing plants to preserve energy for the coiling phase so as to reduce the risk of errors and assure a firm attachment to the support. In this sense, the accuracy trade-off of plants may be mediated by the consumption of energy.

The reduction of movement velocity may also serve to lengthen the time window within which tendrils establish contact points with the support. Previous literature has shown that climbing plants modify strategically contact points when twining around supports of different diameters [34]. Therefore, the extra time needed to reach a thicker support may allow climbing plants to correct tendrils trajectories and select more accurately contact points in order to twine more firmly the support. Indeed, when plants have no other choice but climbing a thick support, they could have the necessity to slow down the movement so as to accumulate more evidence about its the physical characteristics and implement corrective adjustments to reduce the scatter of tendrils end-position. Overall, these data support the hypothesis that despite plants and animals have two very unique evolutionary adaptations for multicellular life, each depending on unique kingdom-specific sets of cells, tissues and organs, they might have evolved signaling networks and mechanisms based on a common toolset from our unicellular common ancestor.

## 5. Conclusions

In conclusion, here we describe a kinematical system solution for the investigation of the mechanisms underlying the movements of plants. This described approach has proved sensitive enough to provide key information regarding various aspects which drive circumnutation patterns. Furthermore, the system defines the framework for a systematic investigation on plant’s nutation. The very fact that it allows to extract key parameters allowing a comparison between plants and animals in terms of movement planning and control makes it a valuable tool for comparative biology.

## Figures and Tables

**Figure 1 animals-11-01854-f001:**
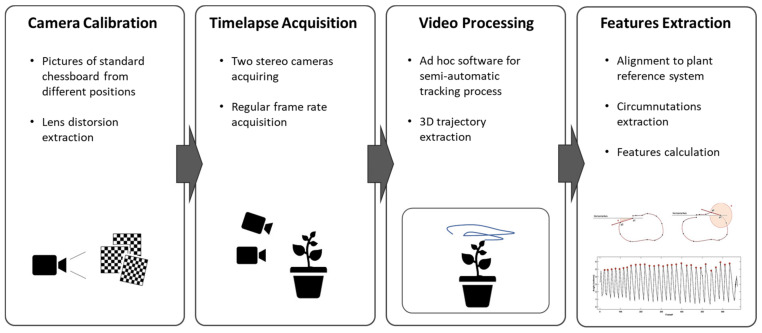
Sequence of the steps for 3D reconstruction and processing of the movement of the plant.

**Figure 2 animals-11-01854-f002:**
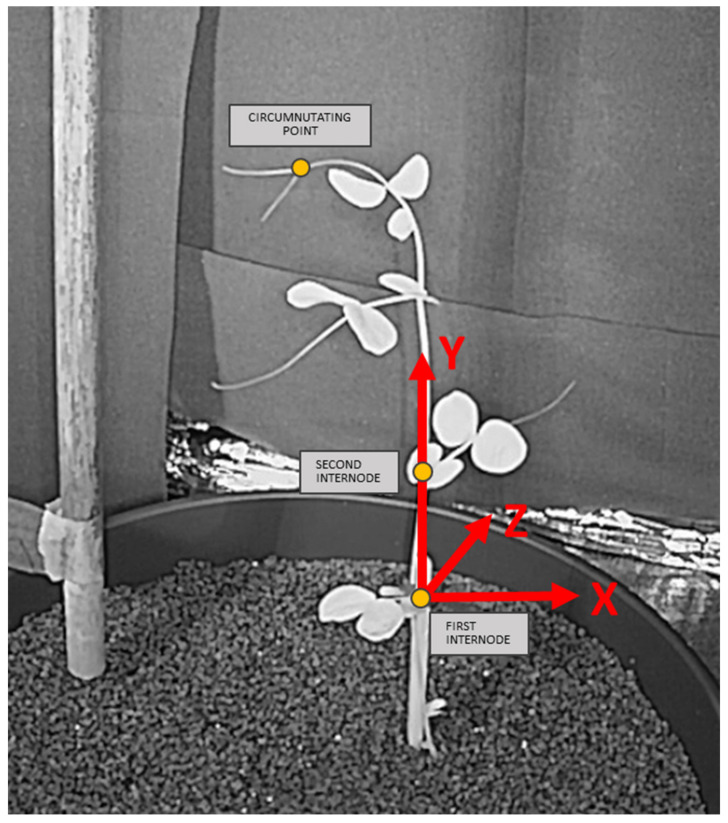
Plant reference system. The y-axis corresponds to the plant vertical axis and the x-axis and z-axis describe the perpendicular planes. The picture shows a pea plant (*P. sativum*) where the vertical axis has been identified as the line passing through the first and the second internode, with the first internode being the origin.

**Figure 3 animals-11-01854-f003:**
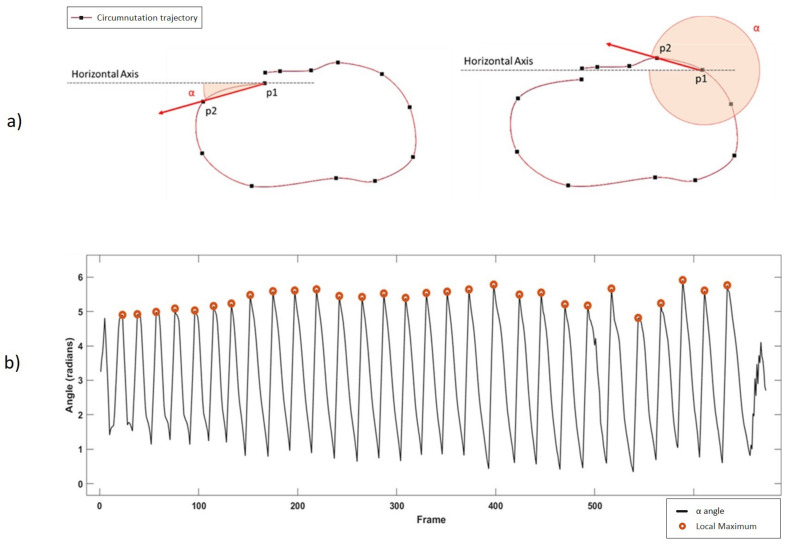
(**a**) α angle computed as the angle between the horizontal axis and the vector of movement between consecutive frames. The movement vector is identified as the vector between the point position in a specific frame (p1) and the point position in the next frame (p2). Movement vector is shown in red for two sample cases: between first and second frame (left) and between ninth and tenth frame (right). (**b**) α angle along the whole movement after a five-sample moving average application. Red dots represent local maxima, two consecutive local maxima encompass a single circumnutation.

**Figure 4 animals-11-01854-f004:**
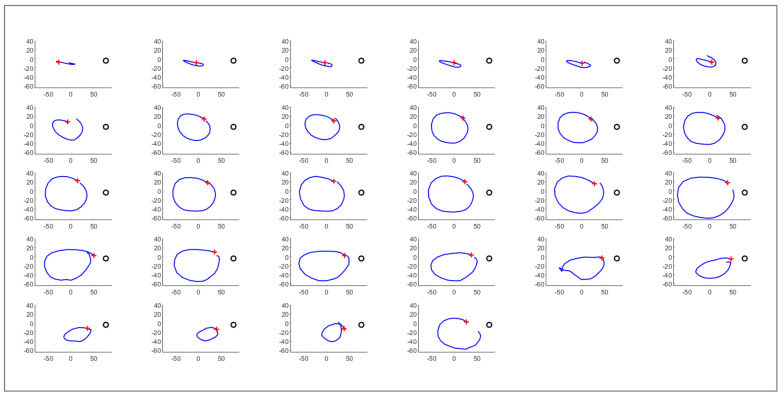
All circumnutations extracted from a complete trajectory of the circumnutating point shown in Figure 2 on the X–Z plane after 35 h of movement. In each box, the trajectory of the tracked point is shown for an identified circumnutation. A red cross represents the starting point of the circumnutation. A black circle represents the support. The reported measurement units of all axes are in millimeters.

**Figure 5 animals-11-01854-f005:**
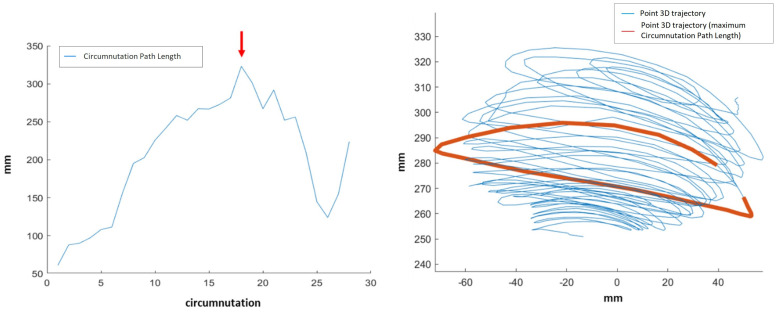
Left: value of circumnutation path length along the circumnutations. Each point represents the 3D path travelled by the circumnutating point during a circumnutation. The red arrow points at the maximum value. Right: circumnutation with maximum value of circumnutation path length displayed in red inside the whole movement trajectory.

**Figure 6 animals-11-01854-f006:**
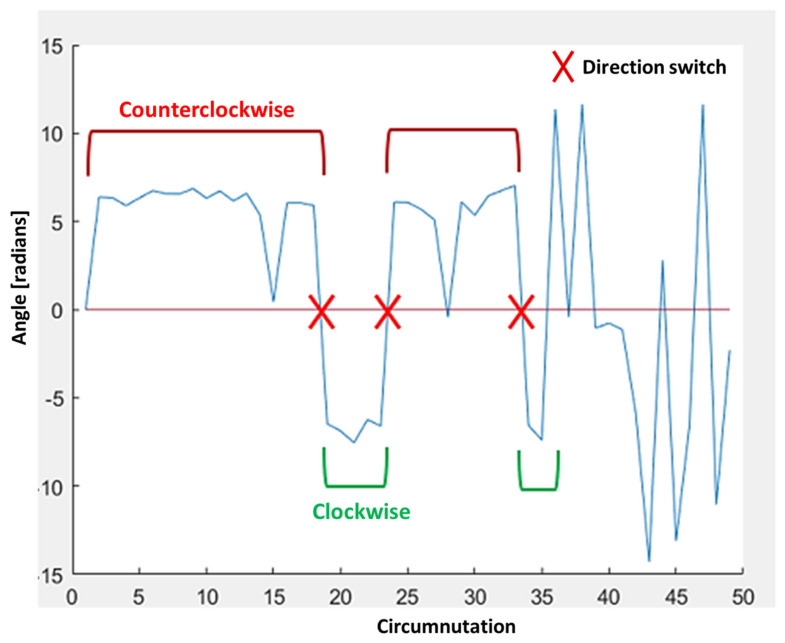
Example of trajectory with three direction switches. Counterclockwise circumnutations are highlighted with red brackets, whereas clockwise circumnutations are highlighted with green brackets. No brackets mean no main direction identified. Red crosses represent changes in the direction of the switches (from counterclockwise to clockwise and vice versa).

**Figure 7 animals-11-01854-f007:**
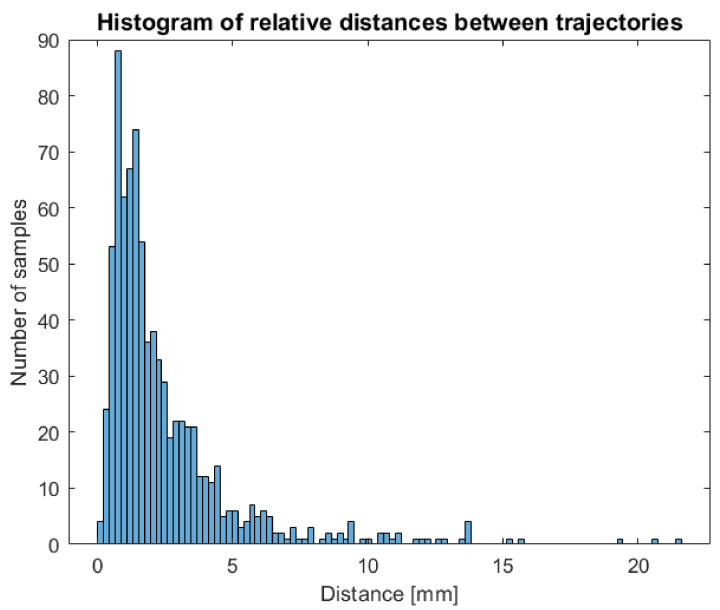
Histogram of relative distances between trajectories obtained from two different users showing that main differences between points are in the range between 0 and 5 mm.

**Figure 8 animals-11-01854-f008:**
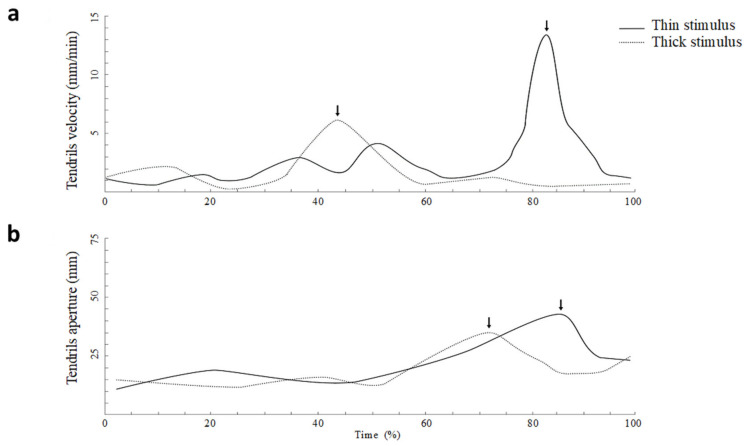
The tendrils’ kinematics was scaled with respect to the size of the supports. Velocity (**a**) and tendrils aperture (**b**) profiles for the movements performed toward either the thick or the thin supports. Arrows indicate the occurrence of maximum peak velocity and maximum grip aperture depending on the thickness of the supports. Please note that when the support is thicker, the peak velocity is anticipated, and the maximum aperture of the tendrils is reached earlier for the thicker with respect to the thinner support. Reprinted from Guerra et al. [2].

**Table 1 animals-11-01854-t001:** Correlation coefficients obtained for the three axes of the 3D trajectories obtained from different users.

Axis	Correlation Coefficient
x	0.999
y	0.998
z	0.993

**Table 2 animals-11-01854-t002:** ICC values obtained comparing circumnutation related features over 29 circumnutations obtained from the same 3D trajectory extracted from two different users.

Feature	ICC
duration_circumnutation	0.995
x_center_coordinates_XZ	0.947
z_center_coordinates_XZ	0.841
x_centroid_coordinates_XZ	0.937
z_centroid_coordinates_XZ	0.784
center_distance_from_origin	0.55
centroid_distance_from_origin	0.72
length_major_axis	0.998
line_integral	0.999
area_circumnutation	0.999
speed_center_XZ	0.955
speed_centroid_XZ	0.981
min_speed_circumnutations	0.967
max_speed_circumnutations	0.989
mean_speed_circumnutations	0.999
center_distance_from_stimulus	0.836
centroid_distance_from_stimulus	0.853
angle_axis_stimulus	0.862
min_dist_stimulus	0.986

## Data Availability

Data describing 3D trajectory obtained by 2 different users on the same plant video are available here: http://doi.org/10.5281/zenodo.5006765.

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
