# Peer review of "Can Plants Move Like Animals? A Three-Dimensional Stereovision Analysis of Movement in Plants"

_animals, 2021, doi:10.3390/ani11071854_

Round 1
Reviewer 1 Report
Review on manuscript
Enabling plants’ behavioral analysis: a stereovision-based approach to kinematics
Valentina Simonetti, Maria Bulgheroni, Silvia Guerra, Alessandro Peressotti, Francesca Peressotti, Walter Baccinelli, Francesco Ceccarini, Bianca Bonato and Qiuran Wang and Umberto Castiello
Summary: The authors present an advanced methodology to study circumnutation of plant stems. The set-up consists of two cameras used to obtain a stereoscopic vision by day and night (infrared recording). Reconstruction of the 3D trajectories can describe various details of the plant movement.
General comments: I am surprised that the manuscript was submitted to the journal "animals" because the study presented analyzes plant movement. mdpi has many other journals where the paper would fit better in terms of cope. I am not a native speaker, but it strikes me that the genetiv s is often used for plants. From my point of view, it should be “movements of plants” and not “plants' movement”.
The structure of the manuscript is confusing. The sections "Material and Methods" and "Results" refer to the presentation of the new method. The section "Discussion" suddenly experiments with the pea plants are described and results are presented. In a completely unscientific manner, conclusions about the underlying mechanisms in the plants are drawn from the motion analyses. The new results of the plants exposed to different various stimulations are almost lost in the “Discussion” section.
In summary, the method describes movement, but does not investigate the underlying physiological or mechanical mechanisms of plant movement. And it is at these points that the manuscript is scientifically weak to incorrect. The term "motor" is completely irritating; certainly not a term in botany, but rather from the engineering sciences. A plant has certainly no "motor". The same applies to the use of the term "cognitive" (see comment (2)). A plant has certainly no central control, no plan and also no brain, which would be necessary for it. I believe that the authors did not consult a botanist and wanted to be a bit smart by using inadequate terminology. This problem can certainly be easily solved if a plant physiologist is consulted.
Title:
(1) The term "behavior" seems to me to belong more to zoology and is associated with an "own will" of animals or humans. This is certainly not the case with plants. I suggest that you include the terms circumnutation or climbing plants in the title instead, which would also tell the reader more precisely what you are presenting in the article.
Abstract:
(2) Line 25: The term "cognitive" is a linguistic anthropomorphisation of plant movement with the attribution of a function that they could not have developed. I would prefer if you could find another term that takes into account that such movements have evolved in the course of evolution, leading to a selective advantage and have therefore prevailed in the descendant generations. According to the literature, cognition refers to "the mental action or process of acquiring knowledge and understanding through thought, experience, and the senses”, and plants have neither a central control nor a brain, the term "cognitive" is excluded in any form. In this form and from the point of view of a botanist, this conclusion can never be drawn.
Keywords:
(3) Plant behavior: see comment 1
(4) Motor control: ??
Introduction:
(5) Line 47: Tipping error in “circumnutation”
(6) Line 48: please cite correctly: “… be modified for the good of the plant”.
(7) Line 78: The full stop is missing after the word mechanism.
Material and methods:
(8) Please note that botanical nomenclature requires that plant species names be written in italics, the genus name in upper case, the species name in lower case, and the first describer not in italics. It is quite sufficient if the first describer is named once, namely when the plant is presented for the first time. Line 169, 174, 279 …: Pisum sativum L. ; Line 185: Pisum sativum
(9) Line 222, 233, 238 …: upper case
(10) Line 293: This information about the exact plant details of the plant (Pisum sativum L. var. saccharatum cv Carouby de Maussane; hereafter P. sativum) must be given in Materials. Later, the plant name may be abbreviated as P. sativum.
(11) Table 1, table 2: Please write the numbers with a dot instead of a comma.
Discussion:
(12) Line 355-357 and 392: Plants have no cognitive process, no intention, no planning and no strategy. Apart from that, the pure description of the movement with the new method does not allow any conclusions to be drawn about the processes in the plant. This sentence is wrong!
(13) Line 374: Citation of publication of the authors team are missing. So ist is not clear whether this are “new” or “old” results.
(14) Line 379: Where does this experiment with the photographs comes from? I would like to hear more about it.
(15) Line 394: What kind of sensorial analysis could you find in plants? I would be interested to hear more about your conclusions or findings.
Conclusion:
(16) Line 396ff: From my viewpoint you solely can describe the movement but you cannot explain the underlying mechanism.
(17) Line 401 and 423ff: see comment (2) and general comments. The comparison of motor planning and controls of plants and animals is a zoomorphic bias.
(18) 404ff: The discussion about the similarities and dissimilarities with Raja et al. is not “Conclusion”. This information should be given in the “Discussion” or “Introduction” section.
Literature:
For further improving your manuscript, please study additional research from biology. For examples the research on nutation of the group of Yasmine Meroz, who have presented a mathematical description of the kinematics of plant nutation based on the interplay between geometry and differential growth (https://doi.org/10.1371/journal.pcbi.1005238, https://doi.org/10.3389/frobt.2020.00089 and others).
Author Response
- ……but it strikes me that the genitive s is often used for plants. From my point of view, it should be “movements of plants” and not “plants' movement”.
R1. As suggested by the reviewer we now adopt “movements of plants”.
- The structure of the manuscript is confusing. The sections "Material and Methods" and "Results" refer to the presentation of the new method. The section "Discussion" suddenly experiments with the pea plants are described and results are presented. In a completely unscientific manner, conclusions about the underlying mechanisms in the plants are drawn from the motion analyses. The new results of the plants exposed to different various stimulations are almost lost in the “Discussion” section.
R2. We agree with the Reviewer that the structure of the manuscript might be misleading. We have applied the following changes:
- The heading ‘Materials and methods’ has been replaced with ‘Methodological aspects’
- The heading ‘Results’ has been replaced with “Testing the methodology”
- Given that the Editor asks to bold out the evidence of similarities across plants and animals movement we now include a new section entitled “Using the method to compare movement planning in plants and animals” in which we show how through our procedures it is possible to reveal that plants and animals exhibit similar kinematical modulations when acting towards objects characterized by a different thickness.
- In summary, the method describes movement, but does not investigate the underlying physiological or mechanical mechanisms of plant movement. And it is at these points that the manuscript is scientifically weak to incorrect. The term "motor" is completely irritating; certainly not a term in botany, but rather from the engineering sciences. A plant has certainly no "motor". The same applies to the use of the term "cognitive" (see comment (2)). A plant has certainly no central control, no plan and also no brain, which would be necessary for it. I believe that the authors did not consult a botanist and wanted to be a bit smart by using inadequate terminology. This problem can certainly be easily solved if a plant physiologist is consulted.
R3. We agree with the reviewer that we are not in the position to investigate both the physiological and mechanical mechanisms of plant movement. Regarding the term “motor”, we now adopt the term “motion”.
- Title:The term "behavior" seems to me to belong more to zoology and is associated with an "own will" of animals or humans. This is certainly not the case with plants. I suggest that you include the terms circumnutation or climbing plants in the title instead, which would also tell the reader more precisely what you are presenting in the article.
R4. The title has been modified. Nevertheless we would like to stress the fact that definitions of ‘behavior’ vary. Intuitively we think of these terms as referring exclusively to humans, but essentially they describe the ability of organisms to respond to environmental challenges in such a way as to optimize fitness. As the Reviewer is certainly aware of, plants have similar properties. Because plants are sessile, they have to cope with various types of biotic and abiotic stresses in their environment, and possess elaborate, dynamic mechanisms to adjust their growth and development accordingly. Plants are as adept as animals and humans in reacting effectively to their ever-changing environment. Of necessity, their sessile nature requires specific adaptations, but their cells possess a network-type communication system with emerging properties at the level of the organ or entire plant. The specific adjustments in growth and development of plants can be taken to represent behavior.
R4. The title has been changed according to the Reviewer’s suggestion.
- Abstract:Line 25: The term "cognitive" is a linguistic anthropomorphisation of plant movement with the attribution of a function that they could not have developed. I would prefer if you could find another term that takes into account that such movements have evolved in the course of evolution, leading to a selective advantage and have therefore prevailed in the descendant generations. According to the literature, cognition refers to "the mental action or process of acquiring knowledge and understanding through thought, experience, and the senses”, and plants have neither a central control nor a brain, the term "cognitive" is excluded in any form. In this form and from the point of view of a botanist, this conclusion can never be drawn.
R5. The term cognitive has been removed. But, although including plant behavior within the domain of cognition is considered absurd by you and others (Adams, 2018; Alpi et al., 2007; Taiz et al., 2019) note that in English-speaking universities, plant behavior was treated up until 1935 in psychology texts (Applewhite, 1975; Warden, 1928; Warden, Jenkins, & Warner, 1935; Yerkes, 1913), and the contribution of plants to our understanding of behavior and cognition continues to be acknowledged (Baluška & Levin, 2016; Baluška & Mancuso, 2009; Calvo Garzón & Keijzer, 2011; Castiello, 2020; Cvrcˇková, Žárský, & Markoš, 2016; Gagliano, 2015; Garzón, 2007; Keijzer, 2017; Parise, Gagliano, & Souza, 2020; Segundo- Ortin & Calvo, 2019; Trewavas, 2014, 2016, 2017; van Duijn, 2017). A unitary view that does not separate plants from animals emerges from that body of works. Although the idea that plants may behave in a cognitive way may baffle you, we are genuinely amazed by the complexity of plant responses, that is, by plants’ ability to adapt to an ever-changing environment. Also, evidence is accumulating (in particular from plant physiologists) supporting notions, formerly considered esoteric, that plants can communicate, remember, decide, and even count, all abilities that one would normally call cognitive if they were observed in animals.
- Keywords:
Plant behavior: see comment 1
Motor control: ??
R6. Key words have been changed
- Introduction:
Line 47: Tipping error in “circumnutation”
Line 48: please cite correctly: “… be modified for the good of the plant”.
Line 78: The full stop is missing after the word mechanism.
R7. We thank the Reviewer for spotting these errors, which have been corrected.
- Material and methods:
Please note that botanical nomenclature requires that plant species names be written in italics, the genus name in upper case, the species name in lower case, and the first describer not in italics. It is quite sufficient if the first describer is named once, namely when the plant is presented for the first time. Line 169, 174, 279 …: Pisum sativum L. ; Line 185: Pisum sativum
Line 222, 233, 238 …: upper case
Line 293: This information about the exact plant details of the plant (Pisum sativum L. var. saccharatum cv Carouby de Maussane; hereafter P. sativum) must be given in Materials. Later, the plant name may be abbreviated as P. sativum.
Table 1, table 2: Please write the numbers with a dot instead of a comma.
R8. All done, thank you.
- Discussion: Line 355-357 and 392: Plants have no cognitive process, no intention, no planning and no strategy. Apart from that, the pure description of the movement with the new method does not allow any conclusions to be drawn about the processes in the plant. This sentence is wrong!
R9. The sentence has been changed. Nevertheless, when animals ‘intend’ to do something, they enact their directedness-toward by moving their muscles; when plants ‘intend’ to do something, their intentionality is expressed in modular growth and phenotypic plasticity. Plant and animal behaviors are the outcomes of the goals set in their respective intentional comportments. In phenomenological terms, each type of plant perception expresses a mode of its intentionality: directedness toward the light in photosensitivity, directedness toward sources of heat in thermo-sensitivity, as well as toward (or away from) self and/or other in kin recognition. In each case, it is a matter not only of perceiving but also of interpreting the signals and making decisions in the face of conflicting signals in a non-automatic manner. Intentionality here assumes the more colloquial sense of a deliberate purposeful behavior. Do plants intend to resist herbivores? Do they intend to resist the force of gravity? Do they intend to resist the common stresses they experience? Studies showing individual root systems growing in such a way as to limit resources to competitors certainly seem to imply intention. The communication network of cells and tissues making up an individual plant may be the mechanistic basis of intention in plant behavior. Just as animals, plants seem to gather information about their surroundings, check it out with their internal and external network systems, and make decisions that reconcile their well-being with the environment.
- Line 374: Citation of publication of the authors team are missing. So is not clear whether this are “new” or “old” results.
R10. We report on our previous results.
- Line 379: Where does this experiment with the photographs comes from? I would like to hear more about it.
R11. Please refer to Guerra et al. 2019 (Scientific Reports).
- Line 394: What kind of sensorial analysis could you find in plants? I would be interested to hear more about your conclusions or findings.
R12. Here we are only speculating about the possible sensorial analysis put in place by plants to code for support thickness and modulate kinematics accordingly. This part has been removed from the new version of the manuscript.
- Line 396ff: From my viewpoint you solely can describe the movement but you cannot explain the underlying mechanism.
R13. We fully agree with the Reviewer.
- Line 401 and 423ff: see comment (2) and general comments. The comparison of motor planning and controls of plants and animals is a zoomorphic bias.
R14. We disagree on the Reviewer’s statement as explained in points 4 and 5.
- 404ff: The discussion about the similarities and dissimilarities with Raja et al. is not “Conclusion”. This information should be given in the “Discussion” or “Introduction” section.
R15. Done
- Literature: For further improving your manuscript, please study additional research from biology. For examples the research on nutation of the group of Yasmine Meroz, who have presented a mathematical description of the kinematics of plant nutation based on the interplay between geometry and differential growth (https://doi.org/10.1371/journal.pcbi.1005238, https://doi.org/10.3389/frobt.2020.00089 and others).
R16. We now refer to these articles within the new version of the manuscript.
Reviewer 2 Report
Report on Simonetti et al.
I liked this paper a lot, it has been very carefully and critically performed and describes in necessary detail how its results were acquired. What the paper shows is that climbing plants recognise a support and adjust their movement to acquire it. Circumnutation is very common in plants and in my own view it is used to improve the accuracy of perception by comparing the signal at each extremity of the nutation and thus can determine more accurately the direction in which to go. Oddly enough I think it is probably an evolutionary remnant from the amoebic origin of plants. 1.5 billion years ago plants and animals separated when Paulinella an amoebic protozoon acquired a blue green algal symbiont. Amoeba when faced with noxious conditions makes continual alternate comparisons between two pseudopodia internally of the strength of the unwelcome conditions and by that means is able to determine the direction of a retreat. In reverse it is able to locate good conditions. All this was reported by Jennings in 1906.
I should add that since this is an animal journal one bonus is that it will help indicate to animal scientists that plants do move but in very different time frame to animals. This has always been a problem that zoologists rarely if ever face up to. We ourselves are animals and to observe behaviour requires a change usually within our visual time frame. Visual images are changed every tenth of a second so anything taking ten minutes for example before any movement can be seen will simply be regarded as unmoving and of no interest. Plants are the dominant organisms on this planet. If calculated from a ratio of oxygen (photosynthesis) to carbon dioxide (respiration) they represent more than 99% of life. Time lapse has indicated that plant move but not in our time frame; it has been casually ignored for centuries. To perhaps emphasise the point if a lecture was given at the rate of one word /three minutes no information of any kind would be conveyed and of course the audience would have long gone.
There have been reports certainly in the 19th century where observation indicated that plants could perceive either an object to avoid or to approach. The literature is however scattered and never subject, unsurprisingly, to proper measurements as described here. This paper and the previous one by Guerra et al, (2019) now provide the necessary and critical measurements to demonstrate the truth of the elderly literature; clearly plants are using some kind of visual process and this will have to be gone into in some research detail. This is why as a technique it must be properly reported as laid out here and must be published.
I have some points that need attention and would be grateful if they were attended to.
The end of the abstract states that new cognitive functions of circumnutation have been uncovered. These will have to be specified particularly since the implicztion is of more than one.
I could find no reference to figure 4 in the script and frankly the size of the numeration will lead to it being ignored or simply skated over or require a magnifying glass to see them. Scientists do not have the time to dwell on figures which are not clear.This figure legend also needs a proper statement; it seems to assume that the reader will understand; it took me while to work out what it was about. Where was the pole placed here for example.
Figure 6 again the legend needs a proper statement. I have seen an unpublished video from Stefan Mancuso who showed a reversal of rotation to capture a pole. But he did not provide the necessary details of observation as laid out here.
Finally Runyon et al., Science (2006) 313:1964 provided time lapse vids of Cuscuta movements towards its prey. These say everything that is really needed and I would recommend that these authors think of providing the equivalent.
Author Response
1. The end of the abstract states that new cognitive functions of circumnutation have been uncovered. These will have to be specified particularly since the implication is of more than one.
R1. We have removed the term ‘cognitive’ from the abstract, but we include a new section entitled “Using the method to test for similarities in plants and animals movement” in which we describe a complex behaviour that one would normally call cognitive if observed in animals.
2. I could find no reference to figure 4 in the script and frankly the size of the numeration will lead to it being ignored or simply skated over or require a magnifying glass to see them. Scientists do not have the time to dwell on figures which are not clear. This figure legend also needs a proper statement; it seems to assume that the reader will understand; it took me while to work out what it was about. Where was the pole placed here for example.
R2. We apologize if figure 4 was unclear. We now provide a new version within the present version of the manuscript.
3. Figure 6 again the legend needs a proper statement. I have seen an unpublished video from Stefan Mancuso who showed a reversal of rotation to capture a pole. But he did not provide the necessary details of observation as laid out here.
R3. We now provide a new version of figure 6 with proper statement.
4. Finally Runyon et al., Science (2006) 313:1964 provided time lapse vids of Cuscuta movements towards its prey. These say everything that is really needed and I would recommend that these authors think of providing the equivalent.
R4. Time-lapse videos for pea plants moving towards either thin or thicker stimuli are now provided as supplementary material.
Reviewer 3 Report
This is very interesting manuscript describing new method - stereovision-based approach to plant organ kinematics - which allows precise analysis of plant organ circumnutations.
Authors should just add to their discussion these two relevant papers:
Bastien R, Meroz Y (2016) The kinematics of plant nutation reveals a simple relation between curvature and the orientation of differential growth. PLoS Comput. Biol. 12(12):e1005238.
Gallentine J, Wooten MB, Thielen M, Walker ID, Speck T, Niklas K (2020) Searching and intertwining: climbing plants and GrowBots. Front. Robot. AI. 7:118
Author Response
1. Authors should just add to their discussion these two relevant papers:
Bastien R, Meroz Y (2016) The kinematics of plant nutation reveals a simple relation between curvature and the orientation of differential growth. PLoS Comput. Biol. 12(12):e1005238.
Gallentine J, Wooten MB, Thielen M, Walker ID, Speck T, Niklas K (2020) Searching and intertwining: climbing plants and GrowBots. Front. Robot. AI. 7:118
R1. Done. Thank you for pointing us towards this relevant literature.